# A Simple 3D Cell Culture Method for Studying the Interactions between Human Mesenchymal Stromal/Stem Cells and Patients Derived Glioblastoma [note 1]

**DOI:** 10.3390/cancers15041304

**Published:** 2023-02-18

**Authors:** Lisa Oliver, Arturo Álvarez-Arenas, Céline Salaud, Juan Jiménez-Sanchez, Gabriel F. Calvo, Juan Belmonte-Beitia, Stephanie Blandin, Luciano Vidal, Victor Pérez, Dominique Heymann, François M. Vallette

**Affiliations:** 1Centre de Recherche en Cancérologie et Immunologie Intégrée Nantes Angers, CRCI2NA, UMR1307, Nantes Université INSERM, 44007 Nantes, France; 2Centre Hospitalier-Universitaire (CHU) de Nantes, 44007 Nantes, France; 3Department of Mathematics, MOLAB-Mathematical Oncology Laboratory, University of Castilla-La Mancha, 13071 Ciudad Real, Spain; 4Micropicell Facility-INSERM UMS 016, CNRS 3556, Structure Fédérative de Recherche François Bonamy, 44007 Nantes, France; 5Institut de Recherche en Génie Civil et Mécanique CNRS-UMR6183, Ecole Centrale de Nantes, Nantes Université, 44300 Nantes, France; 6Unité en Sciences Biologiques et Biotechnologies, US2B, Nantes Université, CNRS, UMR 6286, US2B, 44322 Nantes, France; 7Institut de Cancérologie de l’Ouest, 44805 Saint-Herblain, France

**Keywords:** 3D co-cultures, glioblastoma and mesenchymal stromal cells

## Abstract

**Simple Summary:**

Glioblastoma (GBM) remain an incurable disease despite important efforts to find efficient treatments. GBM are heterogeneous, complex, and plastic tumors and as such it is expected that personalized treatments should be the key to therapeutic success. It is thus of the utmost importance to design new methods to mirror the GBM progression and response to treatments for each patient. Our results show that the co-culture of primary cultures derived from GBM patients with cancer associated fibroblasts derived from mesenchymal stem cells exhibit in vitro growth and morphology close to that found in vivo with patients.

**Abstract:**

We have developed a 3D biosphere model using patient-derived cells (PDCs) from glioblastoma (GBM), the major form of primary brain tumors in adult, plus cancer-activated fibroblasts (CAFs), obtained by culturing mesenchymal stem cells with GBM conditioned media. The effect of MSC/CAFs on the proliferation, cell-cell interactions, and response to treatment of PDCs was evaluated. Proliferation in the presence of CAFs was statistically lower but the spheroids formed within the 3D-biosphere were larger. A treatment for 5 days with Temozolomide (TMZ) and irradiation, the standard therapy for GBM, had a marked effect on cell number in monocultures compared to co-cultures and influenced cancer stem cells composition, similar to that observed in GBM patients. Mathematical analyses of spheroids growth and morphology confirm the similarity with GBM patients. We, thus, provide a simple and reproducible method to obtain 3D cultures from patient-derived biopsies and co-cultures with MSC with a near 100% success. This method provides the basis for relevant in vitro functional models for a better comprehension of the role of tumor microenvironment and, for precision and/or personalized medicine, potentially to predict the response to treatments for each GBM patient.

## 1. Introduction

Glioblastoma (GBM) is the deadliest brain tumor and the outcome for these patients is dismal. The standard treatment includes surgical resection followed by a combination of radio- and chemotherapy [1,2]. The prognosis is poor with a median survival of about 15 months, due to the presence of treatment-resistant tumor-initiating cells or glioblastoma stem cells (GSCs) [3], inter- and intra-tumor heterogeneity, and metabolic plasticity [4]. These features are the main reasons for therapeutic failures, as specificity and efficacy of the treatments are not achievable throughout the total population [5].

It has been shown that in two-dimensional (2D) cell culture systems, which are the most common form of cell culture, cells adapt to the new environment by inducing changes at the genetic, transcriptional and protein levels. Recently, patient-derived cells (PDCs) cultured in 3D systems as tumoroids behave more like the native tumor, retaining their intra-tumor heterogeneity [6,7,8]. Animal models, such as the patient-derived xenografts (PDXs) display some important limitations due to cell selection and a non-natural microenvironment, which often constitutes the major part of the tumor [9]. Thus, an interesting alternative would be a simple and reliable culture method that will retain the heterogeneity of the tumor precluding the selection of cells and that could be maintained in vitro with and/or without cells from the tumor microenvironment (TME) for treatment screening purposes.

It has become increasingly evident that cell lines only approximate properties of the tumor. In fact, it is now apparent that the deregulation within the tumor can be best explained when the contributions of and interactions with the microenvironment are taken into account [8,10]. In tissues, tumor cells interact with the extracellular matrix (ECM), non-tumor cells, and soluble factors all present in the TME; and these interactions would both be responsible for the mechanical properties of the cells and contribute to communication between cells. Given this complex mechanical and biochemical interplay, many important biological properties are absent when cells are cultured in 2D cell cultures. It would be more appropriate to use tissue specific matrices; however, the preparation of these matrices show large variations, are time consuming to prepare, and costly. It is probable that the development of in vitro models, which include the TME, could lead to a better understanding of the effects of drugs on tumor cells since there are now many indications that the biomechanical properties, as well as components of the extracellular matrix (ECM), as well as the matrix rigidity, can influence the proliferation and migration of GBM cells [11,12]. 

The principal component of normal brain ECM is hyaluronic acid and a variety of proteoglycans [13]. On the other hand, the composition of the ECM of GBM is different from the normal brain, consisting of many fibrous collagens, which are essential for the activation of signal transduction [14].

The aim of this study was to develop a simple easy technique to construct a three-dimension (3D) GBM patient-derived cell (PDC) model, which could easily support co-cultures of PDCs and stromal cells of the TME to phenocopy the in vivo tumors. This method can thus be used to develop/adapt treatment to the individual characteristics of each GBM patient in the clinical settings.

## 2. Materials and Methods

### 2.1. Materials

Unless stated otherwise, all cell culture material was obtained from Life Technologies (Cergy Pontoise, France) and chemicals were from Sigma-Aldrich (St. Louis, MO, USA).

### 2.2. Cell Culture

After informed consent, tumor samples classified as glioblastoma, based on the World Health Organization criteria, were obtained from patients undergoing surgical intervention at the Department of Neurosurgery at “Centre Hospitalier Universitaire de Nantes” and the “Tumorothèque IRCNA”. Within 4 h after surgical removal, patient-derived cells (PDCs) were recuperated after mechanical dissociation, as described earlier [15]. All procedures involving human participants were in accordance with the ethical standards of the ethic national research committee and with the 1964 Helsinki declaration and its later amendments or comparable ethical standards. At present, 80 fresh samples obtained before treatment have been processed and stored in a biobank. Those used in this study are cited in Appendix A. Primary GBM cells were cultured in defined medium (DMEM/F12 supplemented with 2 mM L-glutamine, N2 and B27 supplement, 2 µg/mL heparin, 20 ng/mL EGF, 40 ng/mL bFGF, 100 U/mL penicillin, and 100 µg/mL streptomycin) in 100 mm dishes. The medium was changed by the removal of about 70% every 2–3 days. All the experiments with primary GBM cells were performed at early passages. Cells were analyzed for mycoplasma regularly.

Bone marrow mesenchymal stem cells (MSCs) were obtained from the “Tumorothèque IRCNA” and cultured in DMEM complemented with 20% heat-inactivated fetal calf serum, 5 ng/mL bFGF, 100 U/mL penicillin, and 100 µg/mL streptomycin in an atmosphere of 5% CO_2_ and 95% humidity at 37 °C. To prepare CAFs: MSCs were cultured in conditioned medium obtained from PDCs and defined medium at a ratio of 30:70 for at least 7 days. FACS analyses of the different CAF populations prepared with different primary culture conditioned media versus fibroblasts using antibodies directed against CD44, CD90, CD10, CD105, and CD107 (markers used to characterized fibroblasts) and showed marked differences in the expression of these markers between the different CAFs tested while all the different CAF populations expressed αSMA (Appendix A).

### 2.3. Biosphere Formation

For the preparation of the different matrices:Alginate and gelatin were dissolved in HBSS (pH 7.0–7.4) at a concentration of 8% and 10% (*w*/*v*), respectively. For the formation of the biospheres, the 500 µL 8% alginate and 500 µL 10% gelatin were mixed and incubated for 1 h at 37 °C.Collagen type 1 (2 mg/mL, Roche Diagnostics, ref. 14009500) was dissolved in 2 mL 0.2% acetic acid, after which 500 µL 1% collagen and 500 µL 8% alginate were mixed and incubated for 1–2 h at 37 °C.500 µL laminin (1.2 mg/mL, Roche Diagnostics, ref. 112432127001) was mixed with 500 µL 8% alginate, then incubated at 37 °C for 1 h.

Once the matrix was ready, 20 µL PDC suspension (4 × 10^6^ cells) was added, and the solution was mixed without forming air bubbles. Using a 200 µL pipette, a single droplet of the alginate-gelatin-cell solution was added into wells of a 48-well plate containing 300 µL 200 mM CaCl_2_. The CaCl_2_ was replaced by 400 µL defined medium and incubated at 37 °C for 30 min, after which the medium was then replaced by fresh medium.

The culture medium was replaced every 2–3 days over 21 days, after which the cells were treated with 100 µM Temozolomide (TMZ) for 96 h. γ-Irradiation was carried out in a Faxitron CP160 irradiator (Faxitron X-ray Corporation, Villepinte, France) at a dose rate of 1.48 Gy/minute.

To determine cell proliferation, a single biosphere was dissociated by incubation for 3 min incubation in 100 mM Na-Citrate. The cell number and viability were determined using the Countess II automated cell counter (Life Technologies, Courtaboeuf, France). The cells were mixed with Trypan blue (1:1) and loaded into a Countess chamber slide. The image analysis software analyzed the acquired cell images to determine the cell count and viability.

Cell viability was assayed by MTT using a cell viability kit from Abcam (France). MTT 0,5mg/mL was added to the culture and after 4 h at 37 °C, the medium was removed, and formazan precipitates were dissolved in DMSO. The optical density was read at 570 nm on a microplate reader (Molecular Device, San Jose, CA, USA).

To analyze the morphology and to determine the length, area, and circularity of spheroids in the biospheres, images were obtained from five areas in a single biosphere from a minimum of 10 biospheres per condition using a Zeiss microscope (Axio Observer and ZEN 2 program, Axio Observer, Carl Zeiss, Rueil Malmaison, France). The images obtained were analyzed using the FIJI program. Circularity, which is a measure that provides values approaching 1 when a 2D object is close to a circular shape and approaching 0 when it is highly irregular, is defined as:$$Circ=4\pi \frac{Area}{Perimeter^{2}}$$

### 2.4. Determination of Percentage of Tumor Initiating Cells

Cells obtained either from biospheres or from 2D cultures were cultured in CellTak (Life Technologies) coated QIAscout 12,000-microraft plates (Qiagen, Courtaboeuf, France) for 24 h and then 200 rafts containing single cells were recuperated into 96-well plates. After 20 days, the percentage of cells capable of forming colonies were determined.

### 2.5. Spheroid Formation in Biospheres

To analyze the formation of spheroids within the biospheres; whole biospheres were fixed for 30 min with 4% paraformaldehyde in 50 mM CaCl_2_-HBSS, then permeabilized with 0.1% Triton X-100 for 15 min at 37 °C, washed with HBSS, and stained with ActinGreen™ Ready Probe^®^ reagent (Life Technologies) for 30 min at 37 °C in the dark. After washing, the biospheres were counterstained with Hoechst. The biospheres were analyzed using a confocal microscope.

### 2.6. FACS Analysis, Immunocytochemistry and Immunohistochemistry

Biospheres were dissociated manually; cells were recuperated and washed, then incubated 30 min with the primary antibody CD133-APC, CD44-APC, CD10-BV420, or CD90-PE. Data acquisition was performed on a FACS CANTO II (Becton Dickinson, Le Ponbt-de-Claix, France) and analyzed using the FlowLogic software (Miltenyi, Paris, France). To determine the mitochondrial content, the cells were incubated for 30 min with 25 µM MitoTracker Deep Red FM (Life Technologies), a far red-fluorescent dye (abs/em ~644/665 nm), then the data acquisition was performed on a FACS CANTO II (Becton Dickinson) and analyzed using the FlowLogic software (Miltenyi).

For immunocytochemistry, biospheres were fixed with 4% paraformaldehyde for 1 h then permeabilized with 0.1% Triton X-100 for 30 min saturated with 5% BSA and then incubated with rabbit anti-human nestin (Proteintech, Rosemont, IL, USA) and mouse anti-human anti-GFAP (Proteintech). Secondary antibodies coupled to Alexafluor-488 or -568 were added and then the sections were analyzed under a confocal microscope (Nikon A1 Rsi, MicroPicell Facility).

To stain for glycoaminoglycans (GAGs), formaldehyde fixed paraffin-embedded (PPFE) sections of biospheres were colored with Alcian Blue (A5268, Sigma) and then counterstained with Kerechtrot (m00283, Diapath, Martinengo, Italy). The detection of collagen in FFPE sections was done using Masson’s trichrome (F/010210, Microm Microtech, Brignais, France) and counterstained with Weigert’s iron hematoxylin solution. Slides were analyzed after scanning with a Nanozoomer HAMAMATSU (MicroPicell Facility, Nantes Université, France).

### 2.7. Statistical Analysis

Data were analyzed and statistical analyses were performed using GraphPad Prism 7.00 (GraphPad Software, San Diego, CA, USA). Data points are expressed as mean ± SD unless otherwise indicated. * *p* < 0.05, ** *p* < 0.01, *** *p* < 0.001. To study the circularity, the Levene quadratic test was performed. With this test, it is possible to compare the variances for two or more groups, where the null hypothesis assumes all variances to be equal.

## 3. Results

### 3.1. Composition of Biospheres

To determine the optimal cell-free structural material or matrix, 3D biospheres were prepared with PDCs using various different matrices using the protocol described in Figure 1A. Initially, laminin and gelatin were tested and the cell proliferation and the capacity of the cells to form a cellular network were used to validate the matrix The results presented in Figure 1B illustrate that cells proliferated more rapidly in the gelatin than in the laminin-containing matrix. This was observed for two concentrations (0.3–0.6 mg/mL) of laminin while cells proliferated more rapidly in 2% gelatin as compared to 1% gelatin. Microscopic analyses of the cell networks represented in Figure 1C show virtually single cell suspensions at both concentrations of the laminin-containing matrix as compared to gelatin matrix where a network of cell-cell interconnections or spheroids could be distinguished. In addition, PDCs cultured in the presence of gelatin displayed heterogeneous growth both in size and shape. Thus, the proliferation rate and the formation of spheroids suggested that gelatin was a better substrate for the matrix for PDCs than laminin. Further experiments showed that increasing the concentration of gelatin from 1.25 to 5% also increased the proliferation of the cells (Figure 1C,D), without having any effect on the formation of spheroids.

Next, we analyzed the proliferation of PDCs in collagen-containing 3D biospheres at final concentrations of 0.25, 0.5, or 1 mg/mL. As shown in Figure 1E, cells only proliferated in biospheres containing 1 mg/mL collagen and with similar proliferation rates as observed for gelatin-containing biospheres.

### 3.2. Proliferation and Cellular Network in 3D Biosphere

Next, we determined the optimal initial cell concentration required to obtain the most favorable growth. The data in Figure 2A suggest that the initial cell number required to support the proliferation of the PDCs in the biospheres was important. The results show that an initial cell concentration of 4 × 10^4^ cells/biosphere gave an optimal growth over 21 days compared to 2 × 10^4^ or 5 × 10^5^ cells/biosphere. Actually, having too few cells (2 × 10^4^ cells) appeared to delay proliferation and the resulting spheroids formed were of inadequate size to render sufficient cells for detailed post-biosphere analyses before day 21, while having too many (5 × 10^5^ cells) resulted in little or no cell growth (Figure 2A). To validate these results, biospheres in approximately 10 µL were prepared in 1 mg/mL collagen or 5% gelatin, with an initial cell concentration of approximately 4 × 10^4^ cells/biosphere and the data in Figure 2B show that the proliferation in the two types of biospheres was quite similar. Comparisons were performed with additional PDC cultures, and all gave similar results.

Subsequently, cellular networks developed by PDCs in gelatin vs. collagen containing biospheres were assessed over a 3-week period. As depicted in Figure 2C, a unicellular cell suspension was present on day 1 in both types of biospheres. These cells evolved over time into compact oval spheroids. These structures are different from the spheroids/neurospheres observed in 2D-cultures, which were much less compact (Appendix A). To further analyze the cellular structures in the biospheres, the diameters of these spheroids were measured (see experimental section) and, as shown in Figure 2D, there was no statistical difference in spheroid size determined in either gelatin- or collagen-containing matrices.

### 3.3. Cellular Heterogeneity and Interaction Cell-Cell

To determine if there is a change in the phenotype of the cells after 3D biosphere culture, the cells were recuperated after day 36 and phenotypic analyses were carried out by FACS analyses (see Methods section). GBMA1 PDCs (subtype: mesenchymal) grew as loosely associated neurospheres in 2D culture (Appendix A). As shown in Figure 3A, the percentage of cells positive for stemness markers CD133 (proneural), CD90 (mesenchymal), and CD44 (mesenchymal) [16] was similar in 2D culture vs. 3D biospheres with either gelatin or collagen matrices. GBM8 PDC (subtype: proneural) grew as semi-adherent cells in 2D cultures (Appendix A) and showed marked differences in their phenotype in 2D vs. 3D cultures. The data presented in Figure 3B indicated a complete absence of CD90 positive cells and an increase in the number of CD133 positive cells in GBM8 cultured in 3D biospheres. Thus, our 3D model favored the expansion of the proneural subtype but did not alter the mesenchymal subtype.

We also performed colony-forming assays to determine the number of single cells capable of instigating the formation of cell colonies. These colony-initiating cells are considered to be the “cancer initiating cells” or GSCs. For these experiments, GBMA1 cells from 2D and biospheres were plated on micro-raft plates and then rafts containing single cells were recuperated and transferred into wells in 96-well plates. After 20 days the 198 wells from each condition were scored and the percentage of colonies (+10 cells) determined. The data in Figure 3C show that the percentage of colonies formed from cells obtained from 2D was similar to that obtained with cells obtained from 3D biospheres with GBMA1 and GBM 10. These results suggested that there is no modification in the number of GSCs under our 3D culture conditions.

In addition to the ECM, the tumor consists of a heterogeneous cell population that interacts in multiple different ways. Thus, it was important that we could mimic cellular interactions similar to those observed in vivo, imitating the high degree of structural complexity. For this we analyzed the cell-cell interactions, as well as the cellular heterogeneity in biospheres. Whole biospheres prepared with GBM69 PDCs (subtype: mesenchymal) were recuperated after day 21, fixed and labeled for nestin (marker of neural stem cells) and GFAP (marker of glial cells). As seen in Figure 3D cells were labeled with nestin or GFAP and some were labeled with both nestin and GFAP, signifying the presence of a heterogeneous population of cells. Furthermore, the cells appeared elongated with numerous interconnections among the cells inferring an interaction between the cells.

Next, we investigate whether the cells in the spheroids were able to synthesize and secrete an ECM. For this, fixed biospheres embedded in paraffin were sectioned and then stained with Masson’s trichrome to reveal collagen and Alcian blue to stain for glycosaminoglycan (GAGs). The presence of both collagen and GAGs were detected inside and around the spheroids formed in the biospheres (Figure 3E). Collagen and GAGs can interact with a variety of binding partners both within cancer and microenvironment cells and thereby influence cancer progression on multiple levels. The presence of these important components under our conditions indicates that the cells in the tumoroids are capable of synthesis components of the TME, another similarity with in situ tumors.

### 3.4. Mathematical Analysis of Spheroid Growth and Morphology

Several studies have suggested that morphological measures may help to classify and characterize brain tumors [17,18]. To quantify the morphology of the spheroids in the 3D cultures, we determined the circularity of the spheroids formed in the biospheres. It is known that GBM have different GBM molecular subtypes, which display quite different behaviors. To study whether those differences are also observed in 3D cultures, we have used mathematical techniques to analyze each case.

The number of cells was quantified in a single biosphere at different time points, and to avoid other possible factors, the doubling time was estimated over the period during which the cells grew exponentially; consequently, fits were done using the exponential function Equation (2) given below. The resulting doubling times of the different primary cultures can be seen in Figure 4A. Note that the primary cultures GBM3 and GBM8, classified as the proneural subtype, proliferated more slowly compared to the mesenchymal primary cultures GBM22 and GBMA1.

At each time point, the diameter (µm) and the area (µm^2^) of about 200 spheroids within the biospheres were assessed directly from the images, as shown in Figure 4B. With the measurements obtained, the corresponding distribution was reconstructed, and its temporal variation was analyzed in Figure 4C. Our analyses on the dynamics of the diameter and area distributions were based on the following considerations: where $v_{c}$ denotes the characteristic volume of a single cell. The volume of one multicellular spheroid consisting of $n\left(t\right)$ cells at time t is thus given by:(1)$$V_{Tot}\left(t\right)=v_{c}n\left(t\right)$$

If the cells exhibit an exponential growth, then their number at time t would be given by:(2)$$n\left(t\right)=n\left(0\right)e^{\frac{t-t_{0}}{\tau }}$$
where $n\left(0\right)$ is the number of cells at the initial time $t_{0}$ and *τ* denotes the time of cell proliferation. Substituting expression Equation (2) for $n\left(t\right)$ into Equation (1), and taking into account that the initial volume is $V\left(0\right)=v_{c}n\left(0\right)$, we obtain the total spheroid volume:(3)$$V_{Tot}\left(t\right)=V\left(0\right)e^{\frac{t-t_{0}}{\tau }}$$

If the cells adopt a spherical shape during growth, then, since the volume of a sphere with diameter *d* is: $V=\frac{\pi }{6}d^{3}$, substituting into Equation (3), the diameter and the area would evolve according to the following laws:(4)$$d\left(t\right)=d\left(0\right)e^{\frac{t-t_{0}}{3\tau }}$$
(5)$$A\left(t\right)=A\left(0\right)e^{\frac{2\left(t-t_{0}\right)}{3\tau }}$$
where $d_{0}$ and $A_{0}$ are the initial diameter and area of the multicellular structure, respectively.

From the estimated proliferation time *τ* and the initial diameters $d\left(0\right)$ and areas $A\left(0\right)$, measured from the recorded images (see Figure 2D), it is straightforward to compare the experimental diameters and areas at the different time t with the expression for the diameter $d\left(t\right)$ and the area $A\left(t\right)$ that the structures would have if they grew spherically. With this comparison, we determined which primary cultures give rise to structures with different compactness during growth. In Figure 4D,E we show that mesenchymal PDCs (GBM 22) formed compact structures, while proneural PDCs (GBM8) formed fewer compact structures. Similar comparisons were made on several PDCs and the overall results suggest that cells from mesenchymal tumors form more compact cell structures in our biospheres when compared to cell structures observed from proneural tumors under our culture conditions. These results suggest that proneural tumors can adopt very different shapes, while mesenchymal tumors tend to form more uniform spherical structures.

Circularity and sphericity both refer to the same concept but are applied in different dimensions. Circularity (or roundness) measures how close a geometric shape is to a perfect circle in 2D, while sphericity measures how close a 3D volume is to a perfect sphere. Sphericity is calculated as the ratio between the surface of a sphere with the same volume as our 3D volume, and the surface of our 3D volume. The circularity of four different primary cultures was compared among each other showing no significant differences. However, when we compared the circularity of the larger spheroids (in terms of area) to the smaller spheroids in each primary culture, we observed some differences. The variance in the circularity was much larger in the bigger structures in the proneural tumors (Figure 5A,B), with a very high statistical significance (5.7396 × 10^−6^ and 0.00047661) while with mesenchymal tumors (Figure 5C,D), the significance of the differences in variance was much smaller or not significant (0.037342 and 0.38142). These results suggest that proneural tumors can adopt very different shapes, while mesenchymal tumors tend to form more uniform spherical structures.

In order to further analyze tumor compactness, a bigger sample size of a cohort of 340 tumors from the GLIOMAT project was analyzed, as described earlier [19]. Using the concept of sphericity, MRI images from the GLIOMAT were analyzed (Figure 5E). Sphericity was calculated as an approach to estimate their compactness. The cohort was divided into two groups, according to median volume, in order to separate between small and large tumors (threshold volume = 29.09 cm^3^). The sphericity of the two groups was compared using the Mann–Whitney non-parametric test, to determine the difference. The Mann–Whitney test revealed significant differences between the two groups (*p* = 0.000228), with a median sphericity of 0.5892 for the small tumors, and a median sphericity of 0.5455 for the large tumors. Although the magnitude of differences is small, due to large sample size, it is possible to assert that large tumors usually have a significantly smaller sphericity than small tumors. The overall conclusion is that, as tumors grow in size, it is expected that their surface regularity will decrease. These results are in accordance with spheroids in biospheres; as spheroids increase in size, their compactness decreases compared to that of a perfect circle growing at the same rate. This result is in agreement with data obtained by Griveau et al. [20].

### 3.5. Influence of the Tumor Microenvironment Cells on Spheroids in Biospheres

Non-cancerous cells in the TME play an important role in the survival and aggressiveness of GBM [21,22]. To determine whether we could use our model to analyze these intercommunications, CAFs were either isolated from GBM tumors or obtained by culturing normal mesenchymal stem cells (MSCs) with conditioned media obtained from GBM primary cultures for 7 days. These CAFs were then co-cultured with PDCs at a ratio of 1:5. Both types of CAFs cultured in 3D biospheres showed no proliferation (Figure 6A,B). In addition, these cells did not form spheroids in biospheres under our culture conditions, even when cultured at high cell numbers (Figure 6B). To evaluate the effect of the TME on tumor cell growth in biospheres, GFP-labeled CAFs were cultured with PDCs in biospheres and the proliferation was assessed over 3 weeks. GFP-labeled CAFs were visible until day 14 within the biospheres, after which no labeled cells were detected; however, GFP-labeled fragments could be observed within spheroids (Figure 6C). The co-culture of CAFs and PDCs resulted in a significant reduction in overall cell proliferation after day 7 (Figure 6D). However, the mean size of the spheroid within the biosphere was significantly larger in co-cultures as compared to monocultures (Figure 6E) suggesting that there was more cell clustering resulting in larger spheroids in co-cultures. The treatment of these biospheres with 100 μM TMZ for 96 h significantly reduced the number of cells in monoculture biospheres compared to the co-cultures (Figure 6F). These data suggest that the CAFs reduced cell numbers but protected the PDCs from cell death.

### 3.6. Effect of Therapy on Mono- and Co-Cultures of GBM Cells

To determine the effects of radio- and chemotherapy on spheroids, biospheres of mono- and co-cultures of GBMA1 + CAFs were exposed to 50 µM TMZ plus radiation ranging from 2–10 Gy. After 96 h, the cells were recuperated, and the proliferation and viability were determined. The data presented in Figure 7A shows a similar rate of proliferation in the co-cultures from day 20 in all cultures except for the one treated with 50 µM TMZ + 10 Gy, while in the monocultures no viable cells were detected at 50 µM TMZ + 5 Gy and very little proliferation was detected at 50 µM TMZ + 2Gy. We have recently shown that in GBM-CAF co-cultures, the survival of the PDCs was linked to mitochondrial transfer from CAFs to GBM cells [23]. FACS analyses of the mitochondrial content of the cells in the mono- and co-cultures of GBMA1 + CAFs treated with 50 µm TMZ and/or 2 Gy showed that in the absence of CAFs where was a marked reduction in the number of viable mitochondria in PDCs (Figure 7B,C). Interestingly, the number of mitochondria in the PDCs from monocultures treated with both TMZ and radiation was six times lower than control PDCs while either TMZ or radiation induced only a 1.4-fold reduction. We also observed an increase in mitochondria in control PDCs in co-cultures as compared to monocultures. The analysis of the mitochondrial content of the different cultures showed that in the co-cultures, all cell had a healthy population of mitochondria even with a treatment of 50 µM TMZ + 2 Gy.

Biospheres containing GBMG5 cells cultured in the absence or in the presence of CAFs were subjected to three doses of radiation (2 Gy) daily, every 3 days, or weekly. The morphology on day 26 of the spheroids present in mono- and cultures of GBMG5 cells after the different treatments are shown in Figure 7A. The morphology of the spheroids in the monocultures appear ragged with what looks like shedding of the cells from the spheroids, while in the co-cultures the spheroid surface appears smooth.

Next, we looked at CSC markers in mono- and co-cultures of GBMG5 cells after the different regimes of irradiation (Figure 7D,E). Irradiation appears to reduce the number of CD133^+^ cells in the 3 day and weekly irradiated co-cultures while the number of CD133^+^ cells appear to increase in the 3 day and weekly irradiated biospheres. The number of CD133^+^ cells remained the same in daily-irradiated biospheres both in mono- and co-cultures. A similar pattern to CD133^+^ cells was observed for CD44^+^ cells.

## 4. Discussion

Traditional 2D culture systems are very restrictive and fail to recapitulate the original tumor architecture and the TME and do not always allow for the retention the cellular heterogeneity. The consequences of these systems generate a mediocre consistency in assays because in vivo molecular targets are modulated by exchanges between the TME and the tumor cells. These exchanges could include communications that are cell-cell, cell-stroma, as well as cell-matrix. Furthermore, other processes related to tumor construction, including the gradient of O_2_ (formation of hypoxic regions), or nutrients are all lacking in the 2D culture models. Matrix-based 3D culture models are becoming increasingly important tools. The 3D spheroid culture systems permit the development of a complex structure, mimicking the tumor architecture. Indeed, these systems offer the possibility to generate a TME that more closely reproduces that present in the in vivo tumor than the stiff 2D petri dish. The 3D tumor models are crucial to study the influence of the spatial configuration of the cell surface receptors involved in cell-cell as well as cell-TME exchanges. Exploiting a simple 3D biosphere, we have developed a method to embed GBM cells within a cross-linked alginate-gelatin matrix. We have observed that within 7 days of culture, PDCs begin to form multicellular spheroids that increase in size over time. Alginate and gelatin combination have been used as a biocompatible hydrogel matrix to embed cells for the use in 3D bioprinting. The alginate would give viscosity and when cross-linked will afford mechanical support, while gelatin would give elasticity and would promote cell adhesion by being bioactive. Other 3D methods have been used to create multicellular spheroids that use either physical confinement to force the forming of aggregation or the addition of Arg-Gly-Asp (RGD) peptide. The RGD motif is important for the interactions between cells and the ECM and is mediated by cell receptors called integrins; thus, the RGD peptides would act as integrin ligands. It should be noted that the RGD motif is present in gelatin. The composite hydrogel creates a biomimetic environment that facilitates the formation of spheroids without the use of external stresses [24].

It has been shown that chitosan-alginate 3D scaffolds could be used to mimic the glioma microenvironment [25]. We show that our system constitutes an in vitro platform, which more accurately represents the tumor microenvironment for PDCs, as the addition of CAFs, a constituent of tumors, is possible and has an influence on the cancer cell response to treatments. This simple system can be used to understand the interactions between the different components present in the tumor mass and thus to develop new cancer therapeutics.

The mathematical analysis showed that some of the pathophysiological differences between the molecular subtypes that have been observed in GBM patients are also seen in these scaffolds. Mesenchymal tumors are known to have a more proliferative behavior and a worse prognosis, while proneural tumors are more related to an infiltrative or diffusive character. Our 3D biospheres have been able to mimic a very similar behavior, with mesenchymal tumors growing faster and forming compact and spherical structures, while proneural tumors give rise to both less compact and spherical structures, which could be explained due to their diffusive behavior and the tendency to form less cell clustering. To complete the analysis of how fast the different primary cultures proliferate and how compact the resulting spheroids are; the roundness or circularity of the spheroids from different tumors was studied. The circularity of four different primary cultures was compared among each other, showing no significant differences. However, when we compared the circularity of the larger spheroids (in terms of area) to the smaller spheroids in each primary culture, we observed some differences. The variance in the circularity was much larger in the bigger structures in the proneural tumors (GBM3 and 8), with a very high statistical significance (5.7396 × 10^−6^ and 0.00047661), as shown in Figure 4. On the other hand, in the mesenchymal tumors (GBMA1 and 22), the significance of the differences in variance was much smaller or not significant (0.037342 and 0.38142).

In a second set of experiments to analyze the effect of CAFs on the long-term survival of GBM cells after a combined radio- and chemotherapy. The cells in the co-cultures continued to proliferate regardless of the treatment; however, the cells in monocultures only proliferated in cultures treated with 100 µM TMZ, 2 Gy and 5 Gy and no proliferation was detected in biospheres treated with 100 µM TMZ + 2 Gy or 100 µM TMZ + 5 Gy (Figure 7A).

Analysis of the mitochondrial content of the different cultures showed that in the co-cultures, all cells had a healthy population of mitochondria even with a treatment of 100 µM TMZ + 5 Gy. However, in the monocultures, a small population of cells under the control conditions contained no functional mitochondria and this was markedly increased after a treatment of 100 µM TMZ + 2 Gy (Figure 7B). Interestingly, it has been reported that mitochondrial integrity can be correlated with GBM resistance to therapy [26].

FACS analyses of the GSC markers CD133 or CD44 in treated or untreated mono- or co-cultures of GBM + CAFs revealed important differences. In the absence of treatment, for the co-culture with CAFs, both the expression of CD133 and CD44 was increased. However, after radiotherapy, the proportion of CD133^+^ and CD44^+^ cells were reduced when compared to monocultures (Figure 7E). The relation of radio-resistance and the expression of CD44 have been established both in vitro and in patients [27,28]. Thus, our 3D co-cultures exhibit several features observed in treated GBM patients.

## 5. Conclusions

The development of tumoroid technology holds great promises for efficient and easy in vitro testing of new drugs and of new therapeutic approaches. Tumor-derived organoids are 3D structures that closely recapitulate tissue architecture and cancer cells composition.

We have established a protocol which generates spheroids from GBM patients that can be co-cultivated with components of the TME with a remarkable high success (over 90%). We have found that these cultures exhibit gross morphologies and responses to TMZ, which is predictive for patient response to therapy.

The protocol and culture conditions described herein have been made simple in order to provide an easy and reliable clinically relevant model, which could be used to design personalized treatments in the vast majority of GBM patients. This constitutes a basic prerequisite for the use of spheroids for the discovery of efficient therapies in this presently incurable tumor [29,30]. Of note, similar combinations of spheroids and TME cells in 3D conditions have been reported in many different cancers. The major conclusions of these studies have been similar to those reported herein, concluding that the co-culture system in 3D of MSC and cancer cells is promising to evaluate many key functions of cancer cells, such as invasion/metastasis, and demonstrates an important potential to better understand cancer biology and to screen for new therapeutic combinations [31,32,33,34,35,36].

However, one should keep in mind that to reproduce a more complete tumor microenvironment, grafting the co-cultures in animal models will still be a necessary step [37].

## Figures and Tables

**Figure 1 cancers-15-01304-f001:**
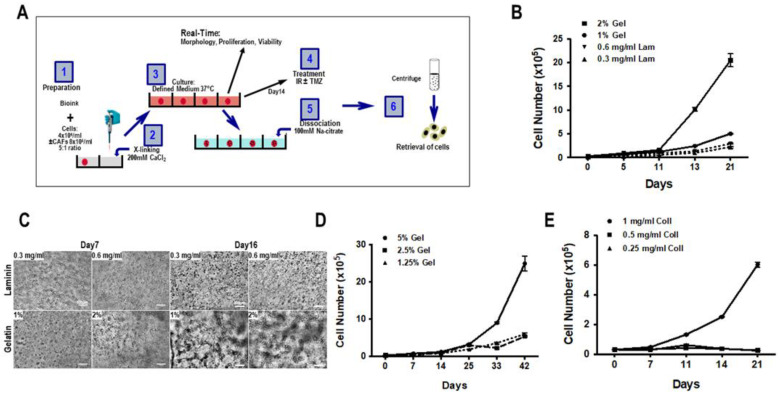
Cell proliferation in 3D biospheres. (**A**) Protocol for the construction of 3D biospheres. (**B**) PDCs (GBM69) were cultured in 3D biospheres composed of either laminin (Lam: 0.3 or 0.6 mg/mL) or gelatin (Gel: 1% or 2%) and cell number was determined over time. This experiment is representative of 3 experiments performed using different PDCs. (**C**) Representative cell morphology of GBM69 cells cultured in biospheres shown in (**B**) was photographed at different times. (**C**) The morphology of the cellular network of cells (GBM69) cultured in gelatin or collagen containing biospheres over time is illustrated. Proliferation of PDCs (GBM71) in biospheres composed either of 1.25%, 2.5%, and 5% gelatin (**D**) or 0.25, 0.5, and 2 mg/mL collagen (Coll). (**E**) Data shown are representative of 3 different primary GBM cultures. Scale bar is equal to100 µm.

**Figure 2 cancers-15-01304-f002:**
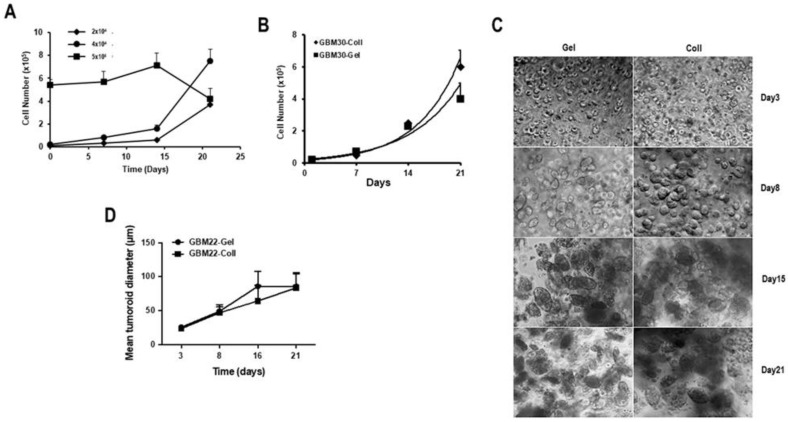
Validation of the composition of matrix. (**A**) Biospheres were made using different initial cell numbers to determine the optimal cell number required to initiate spheroid formation in 3D biospheres. The data presented are those of GBM69 cells but are representative of 3 different PDCs. (**B**) The proliferation of PDCs embedded in either collagen (Coll)- or gelatin (Gel)-containing biospheres was determined over 21 days. The initial cell number was as determined in (**A**) (4 × 10^4^). (**C**) The images of the morphology of the cellular network formed with GBM22 cells cultured in gelatin (Gel) or collagen (Coll) containing biospheres over 3 weeks (scale bar = 50 µm). In total, 30 biospheres were prepared for each condition and 4 representative pictographs were taken of each biosphere. (**D**) Analyses of the mean structure length of the spheroids present in the biospheres over time. The length of the spheroids was determined from the pictographs obtained in (**C**) using Image J, an average of 200 spheroids were measured at each time point. For (**A**–**D**), 4 different PDCs were used and the data presented is that of GBM22 cells.

**Figure 3 cancers-15-01304-f003:**
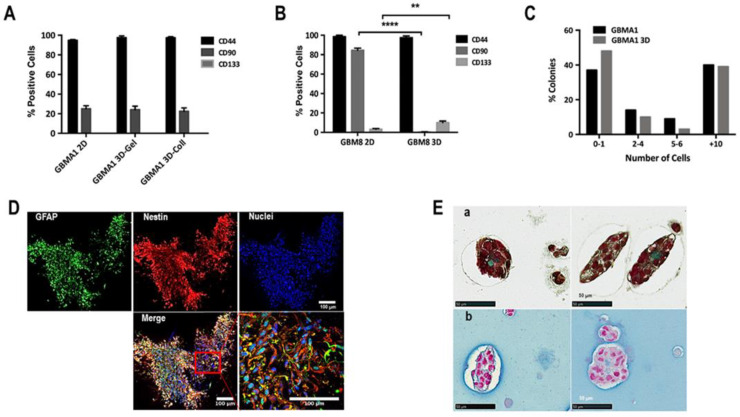
Cell-cell interactions in 3D biospheres. (**A**) FACS analyses of the expression of CD44, CD90, and CD133 in GBMA1 cells (subtype: mesenchymal) obtained from either 2D cultures or 3D biospheres composed of either 2.5% gelatin (Gel) or 1 mg/mL collagen (Coll). (**B**) FACS analyses of the expression of CD44, CD90, and CD133 in GBM8 cells (subtype: proneural) obtained from either 2D cultures or 3D biospheres composed of 2.5% gelatin (Gel). The results are representative of 3 experiments. (**C**) Determination of the percentage of GSCs present in GBMA1 cells obtained from either 2D cultures or 3D biospheres as expressed as the percentage of colonies formed. (**D**) Whole biospheres, obtained on day 21, prepared using GBM69 cells, were labeled for GFAP (green), nestin (red), and nuclei (blue). The insert is a magnification of 5× of a section of the merge and shows tubular interactions between the cells, as well as the double labeling of some cells for both nestin and GFAP (scale bar = 100 µm). (**E**) Histological staining for collagen (**a**) and GAGs (**b**) in FFPE sections of biospheres using GBM3 cells collected after 21 days in culture. Data shown are representative of 3 different primary GBM cultures (scale bar = 50 µm). ** *p* > 0.005; **** *p* > 0.0001.

**Figure 4 cancers-15-01304-f004:**
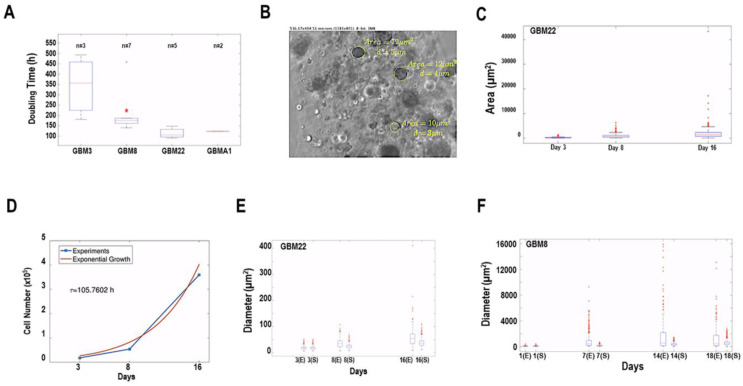
Mathematical determination of cell proliferation and compactness. (**A**) Boxplot of the doubling time of 4 PDC cultures (2 proneural subtypes: GBM3 and GBM8 and 2 mesenchymal subtypes: GBM22 and GBMA1). Where *n* is the number of experiments used to determine the average rate. (**B**) Image from a biosphere showing spheroids and indicating the diameter (µm) and area (µm^2^) of each spheroid measured. (**C**) Boxplots of the distribution of the area of the spheroids measured as in (**C**) for GBM22 on day 3, 8, and 16. (**D**) Graph showing the experimental growth of GBM22 PDCs and the calculated exponential growth required evaluating the estimated proliferation time *τ*. Boxplots of two examples of PDCs in biospheres: (**E**) GBM22 cells which form dense spheroids and (**F**) GBM8 cells which form fewer compact spheroids. Data is presented as areas of the spheroids measured in the biospheres on day 3, 8, and 16 for GBM22 and on day 1, 7, 14, and 18 for GBM8. Two examples of the same PDC culture are represented (E and S). * *p* > 0.01.

**Figure 5 cancers-15-01304-f005:**
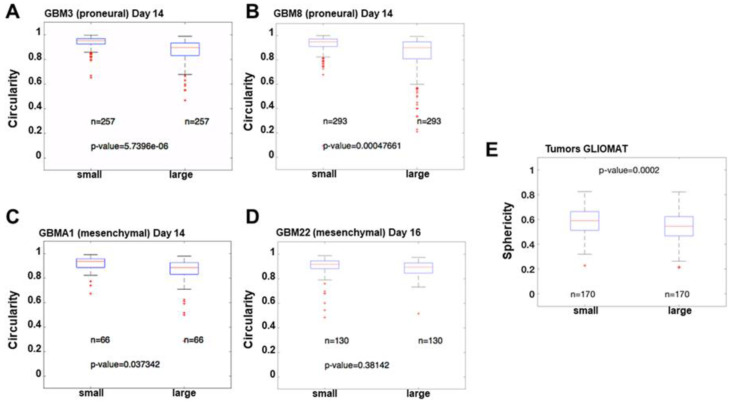
Analysis of circularity. Boxplots of the circularity in small and large structures in biospheres in two proneural PDCs ((**A**,**B**): GBM3 and GBM8) were compared to two mesenchymal PDCs ((**C**,**D**): GBMA1 and GBM22). The data presented show no statistical differences between small and large spheroids in GBMA1 (*p* = 0.037342) and in GBM8 (*p* = 0.38142) biospheres, however, significant differences were present in GBM3 (*p* = 5.7396 × 10^6^) and GBM8 (*p* = 0.00047661) biospheres. Statistical analyses were performed using the Levene quadratic test. (**E**) Boxplots of sphericity in small and large tumors. Data presented show statistical differences between small and large tumors (*p* = 0.000228). Tumor cohort extracted from GLIOMAT project. Tumor volumes were determined from MRI segmentation. red sign statistically different (*p* < 0.01).

**Figure 6 cancers-15-01304-f006:**
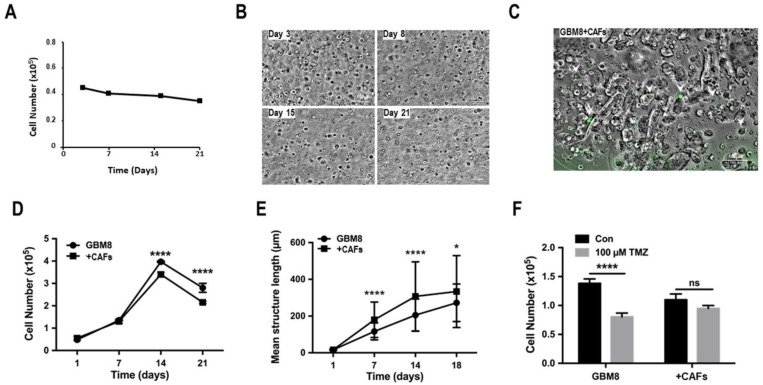
Intercellular exchange with the TME. (**A**) Proliferation of monocultures of CAFs in 3D biospheres. The initial cell number was 4 × 10^4^ cells/biospheres. (**B**) Pictographs depicting the morphology of CAFs cultured in biospheres over 21 days (scale bar = 100 µm). (**C**) Pictograph on day 14 of the cells in co-cultures of GFP-labeled CAFs and PDCs (GBM8) in biospheres (scale bar = 100 µm). Results are representative of the experiment that was done with 5 different PDCs. (**D**) Proliferation of GBM8 cells cultured in the absence or presence of CAFs at a ratio of 5:1 in biospheres. Statistical analyses were done using grouped 2-way ANOVA. (**E**) Analysis of the length of the spheroids formed in biospheres of mono- and co-cultures GBM8 cells and CAFs. Statistical analyses were performed by grouped 2-way ANOVA. (**F**) Quantification of the proliferation in biospheres of mono- and co-cultures GBM8 cells or GBM8 cells plus CAFs treated with 100 µM TMZ for 96 h. Data present are representative of 4 different experiments using different PDCs and/or TASCs. ns = non specific; * significant *p* < 0.01; **** significant *p* < 0.0001.

**Figure 7 cancers-15-01304-f007:**
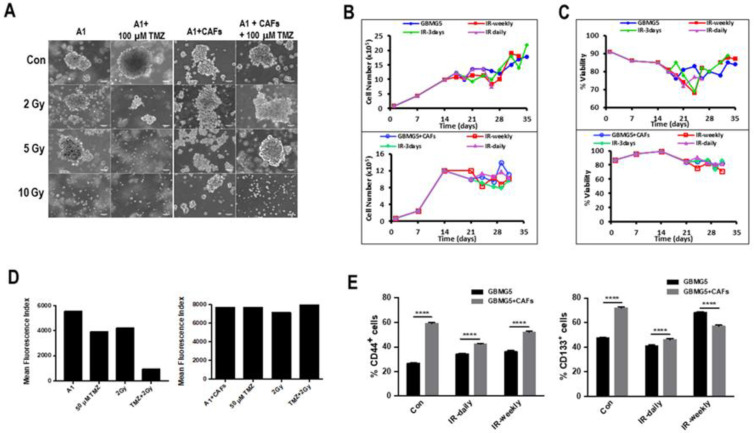
Long-term effects of radiation and TMZ-treatment. (**A**) PDCs from GBMA1 or GBMA1 plus CAFs derived from MSCs were treated with a single dose 100 µM TMZ plus or minus 2 Gy or 5 Gy radiation for 5 days. After 30 days, the morphology and proliferation of the cells was assessed, as described in materials and methods (scale bar = 100 µm). Proliferation (**B**) and viability (**C**) of treated co-cultures. (**D**) Cells obtained from the different cultures in (**A**) were labeled with MitoT Deep-Red and the mean fluorescence index of the mitochondrial mass was determined in the different mono- and co-cultures after FACS analyses. (**E**) FACS analyses to determine the percentage of CD133^+^ or CD44^+^ cells were determined in mono- and co-cultures of GBMG5 cells and CAFs after the different regimes of radiation. **** significant *p* < 0.0001.

## Data Availability

The data presented in this study are available on request from the corresponding author.

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
