# Peer review of "A Simple 3D Cell Culture Method for Studying the Interactions between Human Mesenchymal Stromal/Stem Cells and Patients Derived Glioblastoma"

_cancers, 2023, doi:10.3390/cancers15041304_

Round 1

Reviewer 1 Report (Previous Reviewer 3)

Dear authors 

the manuscript is well designed and experiments are clearly described. the proposed method for the spheroids formation is original and it could be absolutely useful in preclinical studies mimicking the tumor microenvironment better than a 2D system. on the other hand we cannot forget that it is necessary to remember the importance of animal models among preclinical models. I suggest inserting some observations on this subject in the discussion.

Author Response

Dear Editor,

Please find our answer to reviewer 1, we hope this will satisfy this reviewer.

Reviewer 1 comments

the manuscript is well designed and experiments are clearly described. the proposed method for the spheroids formation is original and it could be absolutely useful in preclinical studies mimicking the tumor microenvironment better than a 2D system. on the other hand we cannot forget that it is necessary to remember the importance of animal models among preclinical models. I suggest inserting some observations on this subject in the discussion.

Our answer

We have added a statement about animal models and a new reference  in the conclusion section as suggested by the reviewer.

"However, one should keep in mind that to reproduce a more complete tumor microenvironment, grafting the co-cultures in animal models, will still be a necessary step (37)."

  1. Golebiewska A, Hau AC, Oudin A, Stieber D, Yabo YA, Baus V, Barthelemy V, Klein E, Bougnaud S, Keunen O, Wantz M, Michelucci A, Neirinckx V, Muller A, Kaoma T, Nazarov PV, Azuaje F, De Falco A, Flies B, Richart L, Poovathingal S, Arns T, Grzyb K, Mock A, Herold-Mende C, Steino A, Brown D, May P, Miletic H, Malta TM, Noushmehr H, Kwon YJ, Jahn W, Klink B, Tanner G, Stead LF, Mittelbronn M, Skupin A, Hertel F, Bjerkvig R, Niclou SP. Patient-derived organoids and orthotopic xenografts of primary and recurrent gliomas represent relevant patient avatars for precision oncology. Acta Neuropathol. 2020 Dec;140(6):919-949. doi: 10.1007/s00401-020-02226-7. Epub 2020 Oct 3. PMID: 33009951; PMCID: PMC7666297.

Reviewer 2 Report (New Reviewer)

This manuscript is interesting, but the novelty is not clear. The authors should introduce the 3D cancer-stromal cells research paper and discuss it by comparing these papers.

Major revision should be made. This manuscript would be re-considered only when all the comments were responded.

1. Introduction and Discussion

The authors should introduce the 3D cancer-stromal cells, especially stem cells, research paper and discuss it by comparing these papers. In the current version, it is difficult to understand the strength and publish to Cancers.

The reviewer suggests the papers to be cited.

Review (for concept)

Cancers 202012(10), 2754

Tissue Engineering Part B: Reviews.Jun 2010.351-359.http://doi.org/10.1089/ten.teb.2009.0676

Research papers

Journal of Translational Medicine volume 11, Article number: 28 (2013

https://doi.org/10.1016/j.reth.2021.11.006

https://doi.org/10.1016/j.biomaterials.2013.11.050

https://doi.org/10.1016/j.jconrel.2020.12.054

2.

Cell functions, such as gene expression, should be investigated.

Author Response

Dear Editor,

Please find our answer to reviewer 2. We hope we have successfully answered the reviewer’s comments.

Comments and Suggestions for Authors

This manuscript is interesting, but the novelty is not clear. The authors should introduce the 3D cancer-stromal cells research paper and discuss it by comparing these papers.

Major revision should be made. This manuscript would be re-considered only when all the comments were responded.

  1. Introduction and Discussion

The authors should introduce the 3D cancer-stromal cells, especially stem cells, research paper and discuss it by comparing these papers. In the current version, it is difficult to understand the strength and publish to Cancers.

Our answer: This reviewer has to keep in minds that it is the first demonstration of glioblastoma primary cultures co-cultivated under 3D conditions with mesenchymal stem cells. This has been stated in the introduction.

“The aim of this study was to develop a simple easy technique to construct a three-dimension (3D) GBM patient-derived cell (PDC) model, which could easily support co-cultures of PDCs and stromal cells of the TME to phenocopy the in vivo tumors. This method can thus be used to develop/adapt treatment to the individual characteristics of each GBM patient in the clinical settings.”

We have discussed and added in the conclusion section the references suggested by the reviewer which are describing other types of cancers.

Of note, similar combination of spheroids and TME cells in 3D conditions have been reported in many different cancers. The major conclusions of these studies have been similar to that reported herein, concluding that the co-culture system in 3D of MSC and cancer cells is promising to evaluate many key functions of cancer cells such as invasion/metastasis and demonstrates an important potential to better understand cancer biology and to screen for new therapeutic combinations (31-36).

  1. Nii T, Makino K, Tabata, Y. Three-Dimensional Culture System of Cancer Cells Combined with Biomaterials for Drug Screening. Cancers 2020;12:2754. https://doi.org/10.3390/cancers12102754
  2. Touboul C, Lis R, Al Farsi H, Raynaud CM, Warfa M, Althawadi H, Mery E, Mirshahi M, Rafii A. Mesenchymal stem cells enhance ovarian cancer cell infiltration through IL6 secretion in an amniochorionic membrane based 3D model. J Transl Med. 2013 Jan 31;11:28. doi: 10.1186/1479-5876-11-28. PMID: 23369187; PMCID: PMC3582577.
  3. Nii T, Tabata Y. Immunosuppressive mesenchymal stem cells aggregates incorporating hydrogel microspheres promote an in vitro invasion of cancer cells. Regen Ther. 2021 Dec 10;18:516-522. doi: 10.1016/j.reth.2021.11.006. PMID: 34977285; PMCID: PMC8668441.
  4. Bersini S, Jeon JS, Dubini G, Arrigoni C, Chung S, Charest JL, Moretti M, Kamm RD. A microfluidic 3D in vitro model for specificity of breast cancer metastasis to bone. Biomaterials. 2014 Mar;35(8):2454-61. doi: 10.1016/j.biomaterials.2013.11.050. Epub 2013 Dec 31. PMID: 24388382; PMCID: PMC3905838.
  5. Ferreira LP, Gaspar VM, Monteiro MV, Freitas B, Silva NJO, Mano JF. Screening of dual chemo-photothermal cellular nanotherapies in organotypic breast cancer 3D spheroids. J Control Release. 2021 Mar 10;331:85-102. doi: 10.1016/j.jconrel.2020.12.054. Epub 2021 Jan 1. PMID: 33388341.
  6. Jubelin C, Muñoz-Garcia J, Griscom L, Cochonneau D, Ollivier E, Heymann MF, Vallette FM, Oliver L, Heymann D. Three-dimensional in vitro culture models in oncology research. Cell Biosci. 2022 Sep 11;12(1):155. doi: 10.1186/s13578-022-00887-3. PMID: 36089610; PMCID: PMC9465969.

  1. Cell functions, such as gene expression, should be investigated.

Our answer: The reviewer is right and we currently performing omics analyses of in vitro co-cultures with patient tumors and patients derived xenografts. However, we feel that this work belongs in another publication.

Round 2

Reviewer 2 Report (New Reviewer)

Good revision.

This manuscript is a resubmission of an earlier submission. The following is a list of the peer review reports and author responses from that submission.

Round 1

Reviewer 1 Report

The manuscript by Oliver et al describes a study about how to do 3D cell culture more efficiently for human MSC and PDG. Still, there are some limitations of these methods. Even though in the environment co-culturing tumor spheroids with CAFs, we can’t really set up and detect any other effect from such as immune cells or hormones. However, it is still valuable to suggest these methods for further stem cell spheroids studies for further studies.

There are a few concerns I would like recommend to add or revise. Some statements are pretty confusing.

The Major concerns:

1. The resolutions of every figure are very low so that giving trouble to read clearly. The letters are so small, the symbols (for example, Figure 1B) for each sample group look so similar to each other. Either use different color (even black or various degrees of grey) or clearly different symbols.

2. Every graph requires p-value or n.s.

3. For the material methods, more explanation of CAF culture would need to be added.

4. For the material methods 2.3. Biosphere formation, the alginate, gelatin, collagen and laminin were used for just coating? or they got incubated with the spheroids?

5. In Fig 1c and 2c, the picture resolution should be upgraded and some boundaries for several spheroids (not all of them, just few representative ones) could be added to more visualize clearly.

6. In Fig 2D, GBM doesn’t get bigger than about 80 um diameter size? What about in patient?

7. Every figures used different GBM from different patients. Is there any specific reason?

8. For figure 3, 2D culture method should be added in the Material Methods.

9. What if 2D cultured cells transferred to 3D culture environment the authors showed here? and what about vice versa? Their growth level, any phenotype or any molecular mechanism will be changed?

10. In Fig 3A, there is no CD133 positive rate.

11. After 283-287 lines, the interpretation of the results should be added.

12. In Fig 3c, the explanation of X axis and Y axis is required since the result is confusing.

13. The methods for Figure 4 and 5 are very interesting. These are developed my these authors? Otherewise, the reference should be added.

14. It would be better to add full names for Abbreviations, such as .  

15. Why did you check the circularity? Circularity would be important to check tumor’s growth rate? And in the figure 5, it looks like no significant difference even though these graphs have good p-values. Y axis should be less wide. For example, 0.4-1.0 for Figure 5A, not 0-1.0.  

16. The way the authors obtained the CAFs from GBM tumor in line 401 should be addressed with more details.

17. The statement for Figure 6C (line 408-410) is very confusing. And any protein was dyed with GFP to label CAFs? In Figure 6C, it’s hard to see the GFP.

18. In Fig 6F, the cell numbers are higher in GBM8 control group compared to GBM8+TAFs control group. It looks like GBM8 control group cells grow more. The quantification should be fixed.

19.  In figure 7B, it’s really hard to see the labels. So this 7B is showing that mitochondrial content of cells using FACS analysis? This is confusing.

20. The statements about figure 7c and 7d were missed in the Result.

The minor concerns:

1. It is hard to prove that “PDCs cultured in the presence of gelatin displayed heterogeneous growth both in size and shape” (in Line 196-197) with the experiments here. Only with size and shape difference, could we tell they all have heterogeneous growth?

2. Based on Figure 1, It looks like the higher concentration of gelatin used, the more cells could proliferate. What about any higher than 5% gelatin? They will grow better? Also, what about collagen? The cells will grow better in higher concentration of collagen than 1 mg/ml?

3. In line 412, the proliferation significantly reduced after day 14?

4. Please revise the label for the last 100 uM TMZ in figure 7A.

5. The statements in line 435-437 could be confusing.

6. The full name of IR in Figure 7 should be added in the Figure legend.

7. How they did irradiated biospheres should be mentioned in either Material Methods or Figure legend.

Reviewer 2 Report

The manuscript " A simple 3D cell culture method for studying the interactions between primary Glioblastoma cells and cancer-activated fibroblasts" by Oliver et al. presents development and validation of 3D model based on alginate hydrogels as patient-derived cell carriers for determination of personalized treatment. The idea is interesting and potentially promising for clinical applications.

 Specific comments

Materials and methods

P3L127:

Final concentrations of all components in mixture, as well as concentration of cell in mixture have to be stated here.

P3L129:

The gelation time should be specified.

How did you release cells from spheroids after long term cultivation?

P4L158

"Biospheres were dissociated manually", please explain procedure. In section 2.3 authors said that biospheres were dissociated by Na-Citrate.

 Results

3.1. Composition of biospheres

The mean value with standard deviation of the diameter for each type of biosphere have to be stated here.

Please explain influence of gelatin concentration on proliferation rate, as well collagen. Why the cell proliferate only in presents of 1mg/ml collagen?

Figure 1:

F1A has to be moved to Supplementary Data.

F1C is very small and it is very difficult to notice the cells. I suggest to put this figure as a separate. In addition, in figure caption scale bar has to be stated.

Charts F1B, F1D and F1F are also very small and it is difficult to see what happened at the beginning of cultivation, as well as error bars. Please increase dimensions of all charts.

 P5L217

"The results show that an initial cell concentration of 4x104 cells/biosphere gave an optimal growth over 21 days compared to 2x104 or 5x105 cells/biosphere"

In materials and methods authors said "20 µl PDC suspension (4x106 cells) was added and the solution was mixed without forming air bubbles". Please check final concentration of cell in mixture that you extruded.

 3.2. Proliferation and cellular network in 3D biosphere

Please explain why cells did not proliferate at concentration of 5x105 cells?

Figure 2D. Scale bar has to be stated in figure caption. In addition, insert scale bar on each figure.

 P6L265

"For these experiments GBMA1 cells from 2D and biospheres were plated on micro-raft plates and then rafts containing single cells were recuperated and transferred into wells in 96-well plates. After 20 days the 198 wells from each condition were scored and the percentage of colonies (+10 cells) determined."

This part has to be moved to section Materials and Methods.

 L284

"For this, fixed biospheres embedded in paraffin were sectioned and then stained with Masson’s trichrome to reveal collagen and Alcian blue to stain for glycosa-minoglycan (GAGs)"

This part has to be moved to section Materials and Methods or omitted from Results.

 Fig.3E

Above panel in this figure is not clearly visible.

 Fig 4. is not clearly visible. Please increase dimension of all charts. In addition, scale bar has to be stated in figure caption of Fig4B and insert scale bar in this figure.

 Fig 6

What is abbreviation TAFs? This abbreviation is on figure B, C, D, E and F instead of CAFs, as well as in Supplementary Data. Please check.

 Fig 7. Please put FigA as separate figure in order to make it more visible.

Reviewer 3 Report

The manuscript is very interesting due to possible preclinical application of the model proposed but I have some concerns about the biospheres reproducibility and on their effective ability to reproduce the in vivo model.

1-Did similar bioinks and models based on the use of alginate, gelatin or other describe by others scientists? why and how the authors chose the different substrates/bioinks? in the Introduction the authors say "the composition of the ECM of GBM is different ...consisting of many fibrous collagens" . They should cite others studies to explain why they consider the different bioinks.

2-CaCl2 treatment has been reported in several works. the authors should explain the reason of this method and cite similar application in other studies. 

3-to determine the ability to form spheroids using different bioinks it could be convenient to previously establish the adhesion ability  in 2D ?

4-in the results session the authors compare the ability of spheroids to proliferate, to create cellular network and they compare the cellular heterogeneity. they established that collagen and gelatin give similar results but does not report others information about it in next experiments.

5-The authors investigate and compare cellular heterogeneity of 3D biosphere evaluating the percentage of stemness or proneural markers.

6-they could be evaluate also the EMT transition ability of 2D cultures cells vs. 3D cultures cells.

7- the authors reported assumed a trasfer of mithocondria from CAF to tumor cells of the basis of previuos results. this result should be confirmed by others experiments. for example  by electon microscopy of extracellular particles containing mitochondria in conditioned media or the presence of nanotubes

In general the aim of the present work should be better clarified

Reviewer 4 Report

Authors are ambitious, more details and preciosion are needed. Also descriving images it's auspicable to spent some more details.

Reviewer 5 Report

This manuscript is a methodological work to explore the culture conditions for patient-derived cells (PDCs) in conjunction with mesenchymal stem cells derived cancer associated fibroblasts (CAFs).  Although this work is of interest in the context of personalised medicine, the novelty of this work is limited as other authors have demonstrated the values of 3D cultures and the rationales for this particular study are not properly worked out: 

1. Why were MSCs studied? It is known that other immune cell types such as macrophages are more abundant in GBM. Were experiments conducted to address the effects of these cells as well? If so, please include in a revised version. 

2. why was only proliferation analysed?  A major problem in GBM is the infiltration of the adjacent brain tissue, so what is the effect of CAFs on invasive behaviour?

Besides general concerns, there are some problems associated with this manuscript. 

1. Most figures are difficult to decipher, please use colours for graphs, symbols are too small and the images are blurry.  

2. Scale bars are not explained in Figure legends 

3. Figure 3 E, stainings of Collagen and GAGs is not properly explained, why are there 2 images in a row, colours not explained. No detailed description of staining pattern is provided. 

4. Figure 7 B, C and D lack statistics. 

5. English editing is required and some words accidentally occurring should be removed (e.g. Legend to Figure 3, last word "Data". 

Legend to Figure 3 should be: "Cell-cell interactions in 3D biospheres".